

# Food and light availability induce plastic responses in fire salamander larvae from contrasting environments

Raluca Ioana Băncilă[1,2,*], Florina Stănescu[2,3,4,*], Rodica Plăiaşu[1], Ioana Nae[1], Diana Székely[2,5,6], Sabina E. Vlad[2,4,7] and Dan Cogălniceanu[2,7]

[1] "Emil Racoviţă" Institute of Speleology of Romanian Academy of Sciences, Bucharest, Romania
[2] Faculty of Natural and Agricultural Sciences, Ovidius University Constanţa, Constanţa, Romania
[3] Black Sea Institute for Development and Security Studies, Ovidius University Constanţa, Constanţa, Romania
[4] Center for Research and Development of the Morphological and Genetic Studies of Malignant Pathology, Ovidius University Constanţa, Constanţa, Romania
[5] Museo de Zoología, Universidad Técnica Particular de Loja, Loja, Ecuador
[6] Departamento de Ciencias Biológicas y Agropecuarias, Laboratorio de Ecología Tropical y Servicios Ecosistémicos (EcoSs-Lab), Facultad de Ciencias Exactas y Naturales, Universidad Técnica Particular de Loja, Loja, Ecuador
[7] Chelonia Romania, Bucharest, Romania
[*] These authors contributed equally to this work.

Corresponding authors
Rodica Plăiaşu,
rodica_plaiasu@yahoo.com
Diana Székely, diana@butanescu.com

## ABSTRACT

Phenotypic plasticity has been proposed as a mechanism facilitating the colonisation and adaptation to novel environments, such as caves. However, phenotypic plasticity in subterranean environments remains largely unexplored. Here, we test for plasticity in growth and development of fire salamander larvae (*Salamandra salamandra*) from subterranean and surface habitats, in response to contrasting food availability and light conditions. We hypothesized that: (i) low food availability and absence of light decrease larval growth and delay metamorphosis, (ii) light conditions mediate the effects of food availability on growth and time to metamorphosis, and (iii) larval response to contrasting light and food conditions is shaped by the habitat of origin. Our study showed that reduced food availability significantly delayed metamorphosis and slowed total length and body mass growth rates, while exposure to constant darkness slowed body mass growth rate. However, larvae slowed growth rates and increased time to metamorphosis without compromising size at metamorphosis. The effect of food availability on growth and time to metamorphosis did not change under different light conditions. Fire salamanders from subterranean and surface habitats responded differently only in relation to contrasting food availability conditions. Specifically, larvae from the surface habitat grew faster in high food conditions, while growth in larvae from the subterranean habitat was not influenced by food availability. Initial size also appeared to be an influential factor, since larger and heavier larvae grew slower, metamorphosed faster, and the size advantage was maintained in newly-metamorphosed juveniles. Overall, the results of our experiment suggest that plasticity and local adaptation favor the exploitation of aquatic subterranean habitats for breeding by fire salamanders, allowing successful development even under food shortage and day-length constraints, without compromising metamorphic size. Our findings have implications for conservation because they confirm that phenotypic plasticity plays a critical role in allowing fire salamanders to overcome altered environmental conditions.

## INTRODUCTION

Phenotypic plasticity, *i.e.,* the ability of an organism to change phenotypes in response to environmental conditions, has been proposed as a mechanism of cave colonisation and adaptation to life in cave environments (*Romero, 2009*; *Bilandžija et al., 2020*). Phenotypic plasticity has been shown to evolve in various traits. Two traits known to be particularly plastic are organismal growth (somatic growth) and development (ontogenetic change) (*Relyea, 2001*; *Pfennig et al., 2010*). Organisms with complex life cycles are a particularly good model for studying implications of phenotypic plasticity on trade-offs between growth and development in variable environments. Several environmental factors can trigger plastic responses in growth and development. In surface environments, food availability was found to be the main trigger of plasticity in growth and development in multiple taxa (*Monaghan, 2008*; *Jones et al., 2015*; *Vaissi & Sharifi, 2016*; *Yu & Han, 2020*). Empirical studies suggested that changes in food availability cause a reduction in the amount of energy allocated to somatic growth, resulting in smaller size at metamorphosis and affecting the body condition of individuals (*Enriquez-Urzelai et al., 2013*). Furthermore, the environmental conditions experienced during early developmental stages can be carried over in adult life, with long-term effects on various life-history traits, and ultimately on reproduction and survival, affecting the overall fitness of individuals (*Yoneda & Wright, 2005*; *Székely et al., 2020*). To examine the effects of food availability on growth and development for species that experience metamorphosis, the Wilbur-Collins model was explicitly developed (*Wilbur & Collins, 1973*; *Day & Rowe, 2002*). This model predicts that exposure to poor growth conditions would result in a slower growth rate and a longer developmental period, though a minimum size threshold needs to be attained before undergoing metamorphosis (*Day & Rowe, 2002*). This model also proposes a trade-off between growth and development. Previous studies provided support for a negative relationship between growth and development, where low food availability decreases larval growth rate and size at metamorphosis, and increases the larval period, which allows larvae to attain a threshold size before metamorphosing (*Morey & Reznick, 2000*; *Mueller et al., 2012*).

While the consequences of food availability on growth and development are generally known in surface species that undergo metamorphosis, this knowledge is scarcer for their closely-related counterparts from subterranean environments (*Mejía-Ortíz & López-Mejía, 2005*; *Bilandžija et al., 2020*; *Guillaume et al., 2020*). Subterranean habitats differ from surface ones in several features, that, apart from food scarcity, include limited light or total darkness, and stability of microclimate (*Culver et al., 2004*; *Pipan & Culver, 2012*). The process of colonisation of subterranean habitats (*Pipan & Culver, 2012*) involve exposure to strong selective pressures, and is proposed to be driven by an interplay between phenotypic plasticity and local adaptation, which can allow exploiting habitats with variable and unpredictable environmental conditions (*Romero, 2009*; *Manenti &*

*Ficetola, 2013*). For instance, exposure to contrasting photoperiods can have significant effects on growth and development. Studies on anurans revealed that longer photoperiods can accelerate development, leading to smaller body size at metamorphosis (*Wright et al., 1988*; *Ruchin, 2019*), but overall, the mechanisms regulating photoperiodic responses and their consequences on organism life-history remain largely unknown. Emerging evidence suggests that surface organisms raised in darkness respond by lowering their metabolic rates (*Bilandžija et al., 2020*). Down-regulation of metabolic rates is a mechanism underlying the metabolic depression in response to food deprivation, predicted by the metabolic down-regulation model (*Keys et al., 1950*). Metabolic depression may occur as a physiological adaptation to reduce metabolic costs under specific ecological constraints and may limit growth and development (*Rosen, Volpov & Trites, 2014*; *Rosenfeld et al., 2015*). Effects of food availability and the absence of light on growth and development are unlikely to be independent. To date, no study has explicitly investigated how food availability and light conditions interact to shape the growth and development in subterranean environments.

Knowledge on how the interaction between food availability and light conditions influence larval growth and development may be valuable for amphibian conservation. Amphibians are globally declining at a faster pace than any other vertebrate group (*Stuart et al., 2004*; *Munstermann et al., 2022*). Amphibians occur in both surface and subterranean environments, and for a few species, occasional breeding in subterranean environments has been reported. For example, the fire salamander (*Salamandra salamandra*) typically uses surface streams for breeding, but can also use subterranean waters (*Manenti, Lunghi & Ficetola, 2017*). Various organisms have been reported to move from surface to subterranean habitats to seek refuge from competition, or harsh climate or habitat conditions (*Miaud & Guillaume, 2005*; *Ledesma et al., 2020*). Using subterranean habitats as refugia may also be an efficient strategy to face the current climate changes (*Mammola et al., 2019*). Thus, investigating how amphibian life history traits are shaped by the interaction of various factors relevant to subterranean habitats could be a valuable step towards understanding how these habitats might contribute to the persistence of wild amphibian populations.

In this context, we investigated how the long-term exposure to contrasting food availability and light conditions affects growth and time to metamorphosis in fire salamander larvae. We compared the growth rate and size and time to metamorphosis between larvae that originated from surface and subterranean habitats, when food availability and light conditions were experimentally manipulated. We hypothesized that: (i) low food availability and absence of light decrease larval growth and delay metamorphosis, (ii) light conditions interact with food availability, mediating its effects on growth and time to metamorphosis, and (iii) larval response to contrasting light and food conditions is shaped by the habitat of origin.

## MATERIALS & METHODS

### Sampling and experimental design

We collected 260 *S. salamandra* larvae corresponding to the developmental stage 1 (*Juszczyk & Zakrzewski, 1981*) from three populations in Romania: one inhabiting a surface habitat

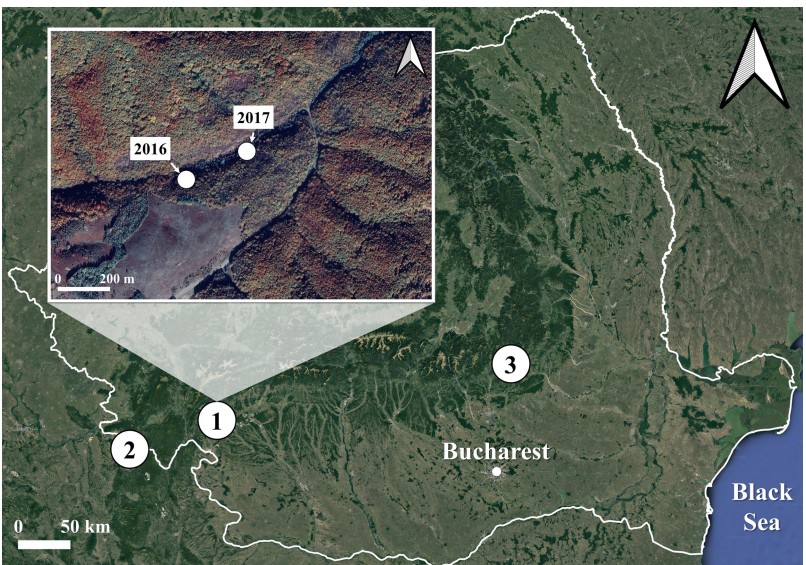

**Figure 1 Map of the sampling sites and occasions for the fire salamander (*Salamandra salamandra*) larvae used in the experiment.** The sites are represented by numbers from 1 to 3. The sampling occasions are shown in the inset map, for the first site, which was sampled both in 2016 and 2017. The other two sites were sampled once, in 2017 only. 1. Iconie (surface stream: 2016, $n = 128$ larvae; 2017, $n = 56$ larvae), 2. Gaura cu Muscă Cave (subterranean habitat: 2017, $n = 47$ larvae), and 3. Buzău Tunnel (subterranean habitat: 2017, $n = 29$ larvae). The white line shows the borders of Romania. The map was based on OpenStreetMap contributors (CC BY-SA 4.0).

and two using subterranean habitats for breeding (Fig. 1). The surface habitat (Iconie, 44.99293°N, 22.78105°E, 325 m a.s.l.) consisted of a deciduous forest crossed by a stream and was sampled on two different occasions: once in April 2016 (Iconie$_{2016}$, $n = 128$ larvae), and once in April 2017 (Iconie$_{2017}$, $n = 56$ larvae). The two populations using subterranean habitats were sampled only once each, in April 2017, from a cave (Gaura cu Muscă, 44.66472°N, 21.69916°E, 100 m a.s.l., $n = 47$ larvae) and a tunnel (Buzău, 45.46981°N, 26.28086°E, 475 m a.s.l., $n = 29$ larvae). The non-permanent nature of the water bodies in the sampled subterranean habitats prevents the larvae from overwintering. The sample size was determined by the availability of stage 1 larvae in natural habitats.

We conducted an experiment with a 2 × 2 factorial design that resulted in four treatments: (1) high food availability and 8-hour light (high-light), (2) low food availability and 8-hour light (low-light), (3) high food availability and 0-hour light (high-dark) and (4) low food availability and 0-hour light (low-dark). The experiment was conducted in the cave-laboratory of "Emil Racoviţă" Institute of Speleology (Cloşani Cave, 45.073064°N, 22.800675°E, 433 m a.s.l.). A thorough description of this cave is provided by *Povară, Drăguşin & Mirea (2019)*. The experimental setup was identical in both years and was distributed among two separate concrete cubicles within the cave: the first cubicle was equipped with a light source which kept the light intensity constant at 50 lx, 8 h a day, while larvae in the second cubicle were kept under constant darkness conditions. The 8-hour photoperiod and 50-lux light intensity were selected to mimic the conditions of natural

shaded habitats at the surface, where salamander larvae are usually found. Humidity and temperature in both cubicles were constant throughout the experiment (close to 100%; $13 \pm 1\,°C$).

The initial wet body mass and body size of larvae were measured (initial size-$TL_0$ and $BM_0$, see below for details) and then the larvae from each population and habitat type were randomly allocated to each of the experimental treatments (Table S1), but with the additional constraint that each treatment had an equal number of larvae (*i.e.,* 65 larvae per treatment). The larvae were housed individually in plastic containers (length × width × height: $21 \times 14 \times 9.5$ cm) with a unique ID, with two cm water depth, following *Warburg (2009)*. All larvae were fed live *Tubifex* sp. every third day and the water was changed after each feeding event. The larvae in high food availability treatments were fed six live prey items per feeding throughout the first 3 weeks, and ten prey items per feeding from the fourth week. The larvae in low food treatments received half the amount of food compared to larvae in the high food treatments.

The effects of food availability and light conditions on larva size were determined based on measurements of individual total length, *i.e.,* the length from the tip of the snout to the tip of the tail (TL) (0.01 mm accuracy), and wet body mass (BM) (0.01 g accuracy). Total length measurements were made from photographs taken with a digital camera (Nikon D3300), using the software ImageJ v. 1.50i (*Schneider, Rasband & Eliceiri, 2012*). Measurements were calibrated using a standardized scale present in each digital image. Wet BM was measured with an electronic Pesola scale, after the larvae were blotted on a wet paper towel to remove excess water (*Beachy, 1995*). Both TL and BM were measured once at the beginning of the experiment ($TL_0$ and $BM_0$), and once at metamorphosis ($TL_{met}$ and $BM_{met}$).

The experiment commenced in April and was terminated in October (after 170 days), both in 2016 and 2017. For each individual, we recorded the time needed to reach metamorphosis ($Time_{met}$) as a measure of developmental rate, defined as the time (days) necessary for the complete resorption of gills and tail fin. Over the 170-day experimental period, 75% of larvae (195 of 260) reached metamorphosis, the fastest after 67 days and the slowest after 169 days. The number of larvae that successfully reached metamorphosis in each treatment was the following: high-dark: subterranean $n = 17$; surface $n = 32$; low-dark: subterranean $n = 16$; surface $n = 30$; high-light: subterranean $n = 14$; surface $n = 35$; low-light: subterranean $n = 15$; surface $n = 36$. The 170-day timeframe was mainly dictated by logistical constraints related to maintaining the larvae; we considered that the threshold of 75% metamorphosed individuals was sufficient for the data analyses. The 65 larvae which did not metamorphose within the 170-days time frame were excluded from the analyses. All larvae and metamorphosed individuals were released in their original habitats at the end of the experiment. There was no mortality in our study.

The collection and rearing of the larvae complied with the Directive 2010/63/UE of the European Parliament and of the Council of 22 September 2010 on the protection of animals used in scientific purposes. Our research was carried out under permit no. 250/20.04.2016 obtained from the Administration of the Domogled - Valea Cernei National Park and permit no. 78/10.02.2016 from the Speleological Heritage Commission.

## Data analyses

All analyses were performed with an *a priori* level of significance of 0.05, in R version 3.6.0 (*R Core Team, 2019*). The dataset comprising the 195 larvae used for the analyses is available as a supplementary file (Table S2). The two samples collected in 2016 and 2017 from the surface site were pooled for the analyses. Growth rates in TL and BM (Growth$_{TL}$ and Growth$_{BM}$) were calculated following *Alcobendas, Buckley & Tejedo (2004)* as the difference between size at metamorphosis (TL$_{met}$ and BM$_{met}$) and the initial size (TL$_0$ and BM$_0$), divided by the time to metamorphosis (Time$_{met}$). We tested the data for normality using the Shapiro–Wilk test and chose the subsequent statistical tests accordingly.

To check for differences in the initial size of larvae between habitats or among populations we used Mann–Whitney and Kruskal–Wallis tests. To test the effects of food availability (low *versus* high), photoperiod (darkness *versus* 8-hour light) and habitat of origin (subterranean *versus* surface), and the two-way interactions of these three factors on each response variable, *i.e.,* Growth$_{TL}$, Growth$_{BM}$, Time$_{met}$, TL$_{met}$ and BM$_{met}$ we used linear mixed models (LMMs) or generalised linear mixed models (GLMMs). We included sampling identity (*i.e.,* Iconie$_{2016}$, Iconie$_{2017}$, Buzău and Gaura cu Muscă) as a random effect in the models. We considered the identity of samples collected in 2016 (Iconie$_{2016}$) and 2017 (Iconie$_{2017}$) from the surface habitat separately in order to account for the potential yearly variation caused by differences in biotic (maternal or litter effects) or abiotic factors. We fitted the models with Gaussian, Poisson or Gamma distribution, and an identity or log link function, and evaluated the fit of competing models using the anova function of the "stats" package. Finally, GLMMs with Gamma distribution and identity link were employed for Time$_{met}$, BM$_{met}$ and TL$_{met}$, while LMMs were used for Growth$_{TL}$ and Growth$_{BM}$.

Adult body size may be strongly influenced by size at birth, *i.e.,* initial size, so that larger offspring will result in larger metamorphs and adults (*Alcobendas, Buckley & Tejedo, 2004*). To account for this potential effect we included TL$_0$ and BM$_0$ as covariates in the models. Since TL$_0$ and BM$_0$ were highly correlated (Spearman's rho = 0.784, $p < 0.001$), we included them separately in models with Time$_{met}$ as a response variable. Subsequently, we compared the resulting models to select the best fitting model. In addition, models with TL$_{met}$ and Growth$_{TL}$ as response variables were fitted with TL$_0$, while the ones with BM$_{met}$ and Growth$_{BM}$ as response variables were fitted with BM$_0$ as covariates.

We used the R package "lme4" version 1.1-21 (*Bates et al., 2015*) for fitting the models. We selected the best fitted models based on analysis of variance tests. We used the function ANOVA from the "car" package (*Fox & Weisberg, 2019*) to assess the significance of model predictors based on Wald and likelihood-ratio chi-square tests (type III analysis of deviance). We performed least square means post-hoc pairwise comparisons with Bonferroni correction for multiple tests, with the packages "multcomp" (*Hothorn, Bretz & Westfall, 2008*) and "emmeans" (*Russell, 2021*). All graphics were done with the R package "ggplot2" (*Wickham, 2016*).

## RESULTS

### Initial size and its effects on traits at metamorphosis

The initial size ($BM_0$ and $TL_0$) of the larvae varied significantly across samples (Kruskal–Wallis test, $BM_0$: $H = 31.842$, $df = 3$, $p < 0.001$; $TL_0$: $H = 36.114$, $df = 3$, $p < 0.001$) and between habitat types (Table S3, Mann–Whitney test, $BM_0$: $W = 5232$, $p = 0.002$; $TL_0$: $W = 5723.5$, $p < 0.001$), with larvae from the subterranean habitats being larger compared to those from the surface habitat (mean $\pm$ SE, subterranean: $BM_0 = 0.315 \pm 0.015$ g; $TL_0 = 38.73 \pm 0.66$ mm; surface: $BM_0 = 0.257 \pm 0.04$ g; $TL_0 = 35.79 \pm 0.27$ mm). However, the initial size of the larvae did not differ significantly across treatments (Kruskal–Wallis test, $BM_0$: $H = 2.2$ 09, $df = 3$, $p = 0.530$; $TL_0$: $H = 2.466$, $df = 3$, $p = 0.481$, Fig. S1).

Growth rates were influenced significantly by the initial size (Table 1), larger and heavier larvae exhibiting lower growth rates in terms of both total length and body mass, respectively (Table 1, Fig. 2). Time to metamorphosis was best explained by a model that included $BM_0$ rather than $TL_0$. $BM_0$ had a significant negative effect on time to metamorphosis, thus larvae with a lower $BM_0$ attained metamorphosis later (Table 1, Fig. 3). Total length at metamorphosis was significantly affected by $TL_0$, larger larvae attaining larger total lengths at metamorphosis; there was no significant effect of the initial body mass on body mass at metamorphosis (Table 1).

### Hypothesis 1: low food availability and absence of light decrease larval growth and delay metamorphosis

We found a significant main effect of food availability on only one of the five metamorphosis traits studied, specifically time to metamorphosis (Table 1). Larvae raised in low food availability treatments reached metamorphosis significantly later compared to those raised with high food availability (mean $\pm$ SE, low = $118.2 \pm 3.03$ days, high = $96.6 \pm 1.90$ days, $p = 0.003$; Fig. 4). Regarding photoperiod, we found a significant main effect on body mass growth rate (Table 1). Thus, larvae raised in darkness exhibited a significantly slower growth rate compared to those raised in the 8-hour photoperiod treatments (mean $\pm$ SE, dark = $0.0044 \pm 0.0001$ g/day, light = $0.0046 \pm 0.0001$ g/day, $p = 0.040$) (Table 1 and Fig. 5).

### Hypothesis 2: light conditions interact with food availability, mediating its effects on growth and time to metamorphosis

The interaction between food availability and photoperiod did not significantly affect either growth rates or time to and size at metamorphosis (Table 1).

### Hypothesis 3: larval response to contrasting light and food conditions is shaped by the habitat of origin

Habitat type and food availability had a significant interactive effect on growth rates (Table 1). Total length growth rate was significantly higher in larvae from the surface habitat raised in high compared to those in low food availability treatment (mean $\pm$ SE, surface-low = $0.116 \pm 0.004$ mm/day, surface-high = $0.155 \pm 0.005$ mm/day, $p < 0.001$, Table 2, Fig. 6A). Instead, larvae from subterranean habitats showed similar total length growth rates between low and high food availability (Table 2, Fig. 6A). Body mass

**Table 1** Results of the best fitting models testing the effects of initial body mass ($BM_0$), initial total length ($TL_0$), habitat (subterranean versus surface), photoperiod (darkness versus 8-hour light) and food availability (low versus high) and their two-way interactions on body mass and total length growth rates ($Growth_{BM}$ and $Growth_{TL}$), time to metamorphosis ($Time_{met}$), and body mass and total length at metamorphosis ($BM_{met}$ and $TL_{met}$). Significant effects are in bold. $\chi^2$, chi-square test; df, degrees of freedom; $p$, level of signification; statistic – test statistic, *i.e.*, $z$ (for generalised mixed modes) or $t$ (for linear mixed models); $R^2$ - effect size.

| Fitted model | $\chi^2$ | df | $p$ | statistic | $R^2$ |
|---|---|---|---|---|---|
| $Growth_{BM}$ ($R^2 = 0.462$) | | | | | |
| **$BM_0$** | **25.405** | **1** | **<0.001** | **−5.04** | **0.124** |
| **Habitat** | **4.761** | **1** | **0.029** | **2.182** | **0.073** |
| **Photoperiod** | **4.201** | **1** | **0.040** | 2.050 | **0.020** |
| Food | 0.198 | 1 | 0.655 | −0.446 | 0.001 |
| **Habitat × Food** | **8.652** | **1** | **0.003** | **−2.941** | **0.041** |
| Habitat × Photoperiod | 0.246 | 1 | 0.619 | −0.497 | 0.001 |
| Photoperiod × Food | 1.961 | 1 | 0.161 | −1.401 | 0.009 |
| $Growth_{TL}$ ($R^2 = 0.724$) | | | | | |
| **$TL_0$** | **124.399** | **1** | **<0.001** | **−11.153** | **0.394** |
| Habitat | 0.878 | 1 | 0.348 | 0.937 | 0.052 |
| Photoperiod | 3.193 | 1 | 0.073 | 1.787 | 0.015 |
| Food | 0.749 | 1 | 0.386 | −0.865 | 0.004 |
| **Habitat × Food** | **6.596** | **1** | **0.010** | **−2.568** | **0.030** |
| Habitat × Photoperiod | 0.187 | 1 | 0.665 | 0.433 | 0.001 |
| Photoperiod × Food | 3.270 | 1 | 0.070 | −1.808 | 0.015 |
| $Time_{met}$ ($R^2 = 0.555$) | | | | | |
| **$BM_0$** | **51.270** | **1** | **<0.001** | **−7.093** | **0.215** |
| Habitat | 1.053 | 1 | 0.304 | −1.020 | 0.019 |
| Photoperiod | 0.057 | 1 | 0.811 | −0.238 | 0.000 |
| **Food** | **8.420** | **1** | **0.003** | **2.900** | **0.032** |
| **Habitat × Food** | **7.753** | **1** | **0.005** | **2.780** | **0.031** |
| Habitat × Photoperiod | 0.012 | 1 | 0.909 | −0.113 | 0.000 |
| Photoperiod × Food | 0.310 | 1 | 0.577 | −0.557 | 0.000 |
| $BM_{met}$ ($R^2 = 0.265$) | | | | | |
| $BM_0$ | 0.136 | 1 | 0.712 | −0.367 | 0.001 |
| Habitat | 0.059 | 1 | 0.807 | 0.243 | 0.014 |
| Photoperiod | 1.447 | 1 | 0.228 | 1.203 | 0.007 |
| Food | 2.284 | 1 | 0.130 | 1.511 | 0.010 |
| Habitat × Food | 0.983 | 1 | 0.321 | −0.991 | 0.004 |
| Habitat × Photoperiod | 0.226 | 1 | 0.634 | −0.476 | 0.001 |
| Photoperiod × Food | 1.371 | 1 | 0.241 | −1.171 | 0.006 |

**Table 1** (*continued*)

| Fitted model | $\chi^2$ | df | $p$ | statistic | $R^2$ |
|---|---|---|---|---|---|
| $TL_{met}$ ($R^2 = 0.468$) | | | | | |
| **$TL_0$** | **7.156** | **1** | **0.007** | **2.675** | **0.032** |
| Habitat | 0.073 | 1 | 0.786 | −0.271 | 0.002 |
| Photoperiod | 2.343 | 1 | 0.125 | 1.531 | 0.010 |
| Food | 0.315 | 1 | 0.574 | 0.562 | 0.001 |
| Habitat × Food | 0.093 | 1 | 0.759 | 0.306 | 0.000 |
| Habitat × Photoperiod | <0.0001 | 1 | 0.999 | 0.001 | 0.000 |
| Photoperiod × Food | 2.595 | 1 | 0.107 | −1.611 | 0.011 |

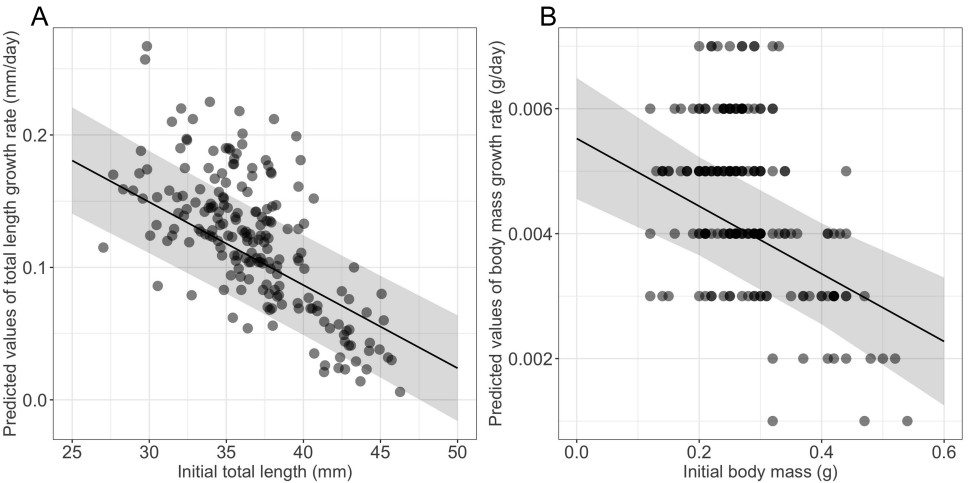

**Figure 2** **Effects of initial size on growth rates in fire salamander (*Salamandra salamandra*) larvae.**
(A) Initial total length effect on the total length growth rate; and (B) initial body mass effect on the body mass growth rate (see also Table 1 for statistical tests). The shaded area represents the 95% confidence intervals.

growth rate was significantly higher in larvae from the surface habitat raised in high food availability treatments (mean ± SE, surface-high = 0.0055 ± 0.0001 g/day) compared to those from subterranean and surface habitats raised in low food availability treatments (mean ± SE, subterranean-low = 0.0035 ± 0.0002 g/day, $p = 0.039$; surface-low = 0.0042 ± 0.0001 g/day, $p < 0.001$, Table 2, Fig. 6B). Habitat type and food availability also had a significant interactive effect on time to metamorphosis (Table 1). Larvae from both surface and subterranean habitats raised in high food availability underwent metamorphosis significantly faster than larvae in low food availability treatments (Table 2, Fig. 7).

The interaction between habitat type and photoperiod did not significantly affect either growth rates or time to metamorphosis (Table 1). However, habitat type had a significant main effect on body mass growth rate (Table 1). Thus, larvae from subterranean habitats increased in body mass significantly slower compared to those from the surface habitat

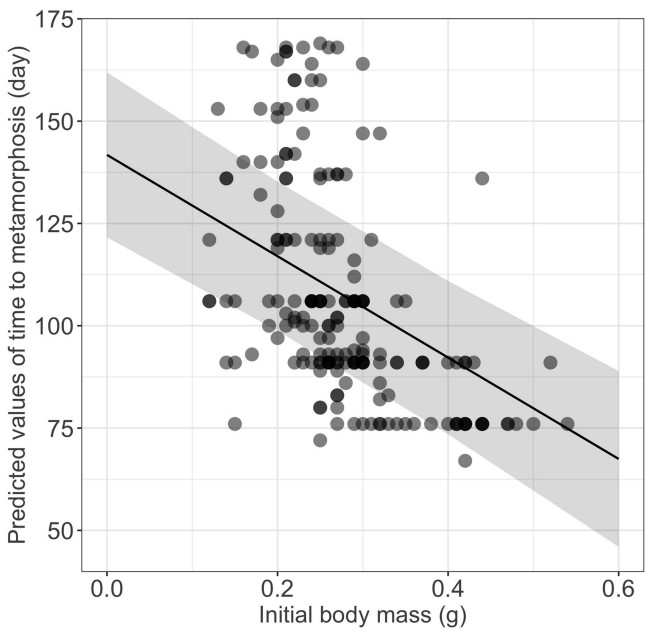

**Figure 3** **Effect of initial body mass on time to metamorphosis in fire salamander (*Salamandra sala-mandra*) larvae.** (See Table 1 for statistical tests). The shaded area represents the 95% confidence intervals.

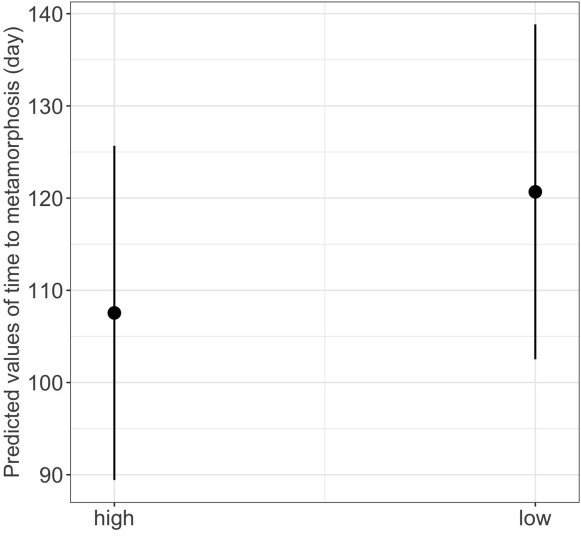

**Figure 4** **Effect of food availability (high *versus* low) on time to metamorphosis in fire salamander (*Salamandra salamandra*) larvae.** (See Table 1 for statistical tests). The black closed circles indicate the mean values of time to metamorphosis and lines represent the 95% confidence intervals.

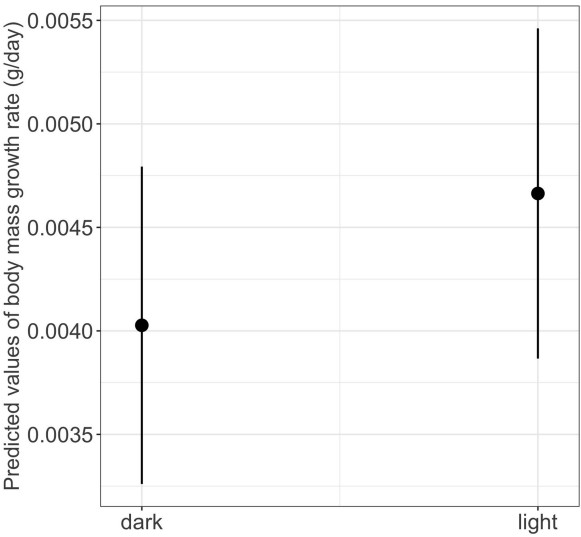

**Figure 5** **Effects of photoperiod on body mass growth rate in fire salamander (*Salamandra salamandra*) larvae.** The photoperiod treatments were Dark (0-hours photoperiod) and Light (8-hours photoperiod). Closed circles and lines represent the mean value and the 95% confidence intervals.

(mean $\pm$ SE, subterranean = 0.0038 $\pm$ 0.0001 g/day, surface = 0.0048 $\pm$ 0.0001 g/day, $p$ = 0.029) (Table 2 and Fig. 8).

Although body mass and total length at metamorphosis were on average higher in larvae from the surface compared to those from the subterranean habitat (mean $\pm$ SE, surface: $BM_{met}$ = 0.79 $\pm$ 0.01 g; $TL_{met}$ = 50.63 $\pm$ 0.34 mm; subterranean: $BM_{met}$ = 0.71 $\pm$ 0.01 g; $TL_{met}$ = 48.23 $\pm$ 0.52 mm), none of the tested main factors (*i.e.*, food availability, photoperiod or habitat type), nor their two-way interaction showed any significant effect on these traits (Table 1, Fig. S2).

## DISCUSSION

Our experimental study demonstrated the importance of phenotypic plasticity and local adaptation in shaping growth and time to metamorphosis in fire salamanders. We found that reduced food availability significantly delayed metamorphosis and differently slowed length and mass growth rates in larvae from subterranean and surface habitats, while darkness slowed mass growth rate only. Importantly, the effects of food availability were not dependent on photoperiod. Despite the slower growth and increased time to metamorphosis, there were no carry-over effects in terms of size at metamorphosis. These findings suggest that plastic responses to environmental stressors may allow individuals to persist in changing environments without compromising their size. Furthermore, we found that the effects of food availability on growth and time to metamorphosis were contrasting in fire salamanders from subterranean and surface habitats, highlighting the role of local adaptations in shaping responses to environmental stressors.

**Table 2 Contrasts of the significant interaction effects of food availability (low versus high) and habitat (subterranean versus surface) on body mass and total length growth rates (Growth$_{BM}$ and Growth$_{TL}$), and time to metamorphosis (Time$_{met}$). Significant effects are in bold.** SE, standard error; statistic, test statistic, *i.e.*, *z* (for generalised mixed modes) or *t* (for linear mixed models); adj. *p*, sequential Bonferroni adjusted significance level.

| | Contrast estimate | SE | statistic | adj. *p* |
|---|---|---|---|---|
| Growth$_{BM}$ | | | | |
| subterranean high - surface high | −0.001049 | 0.00049 | −2.118 | 0.228 |
| subterranean high - subterranean low | 0.0003506 | 0.0002746 | 1.277 | 1.000 |
| subterranean high - surface low | 0.0002768 | 0.0004950 | 0.559 | 1.000 |
| **surface high - subterranean low** | **0.001399** | **0.0004974** | **2.813** | **0.039** |
| **surface high - surface low** | **0.001326** | **0.0001836** | **7.218** | **<0.001** |
| subterranean low - surface low | −0.00007378 | 0.0004981 | −0.148 | 1.000 |
| Growth$_{TL}$ | | | | |
| subterranean high - surface high | −0.026 | 0.026 | −1.017 | 1.000 |
| subterranean high - subterranean low | 0.013 | 0.006 | 1.990 | 0.306 |
| subterranean high - surface low | 0.008 | 0.026 | 0.332 | 1.000 |
| surface high - subterranean low | 0.040 | 0.026 | 1.544 | 0.765 |
| **surface high - surface low** | **0.035** | **0.004** | **7.509** | **<0.001** |
| subterranean low - surface low | −0.005 | 0.026 | −0.196 | 1.000 |
| Time$_{met}$ | | | | |
| subterranean high - surface high | 14.257 | 13.469 | 1.059 | 1.000 |
| **subterranean high - subterranean low** | **−11.811** | **3.893** | **−3.033** | **0.014** |
| subterranean high - surface low | −11.451 | 13.568 | −0.844 | 1.000 |
| surface high - subterranean low | −26.069 | 13.462 | −1.936 | 0.316 |
| **surface high - surface low** | **−25.709** | **3.059** | **−8.404** | **<0.001** |
| subterranean low - surface low | 0.359 | 13.576 | 0.026 | 1.000 |

## Food availability and photoperiod act separately on metamorphosis traits

Previous studies have indicated that food availability affects growth and development in several amphibian species. Most of the studies showed that foraging shifts to higher food quantity or quality results in increased growth, which translates to faster developmental rates (*Leips & Travis, 1994*; *Beachy, 1995*; *O'Laughlin & Harris, 2000*; *Álvarez & Nicieza, 2002*; *Hickerson, Barker & Beachy, 2005*; *Jones et al., 2015*) and/or an increase in size at metamorphosis (*Pandian & Marian, 1985*). A similar study (*Manenti et al., 2023*) on European fire salamander larvae from different altitudes showed that rich food induced higher growth rates and earlier metamorphosis at smaller sizes, and highlighted that both environmental conditions and local adaptations determine the plastic response of larvae. However, it has been shown that acceleration or deceleration of development rate, depends on the developmental stage at which the foraging shift occurs (*e.g.*, *Crump, 1981*; *Alford & Harris, 1988*; *Newman, 1994*; *Tejedo & Reques, 1994*; *Álvarez & Nicieza, 2002*). According to the Wilbur-Collins model, in the case of deceleration of development rate, the predictions are that larvae delay the metamorphosis either to attain a critical size at metamorphosis or to capitalize on the rapid growth opportunity (*Wilbur & Collins,*

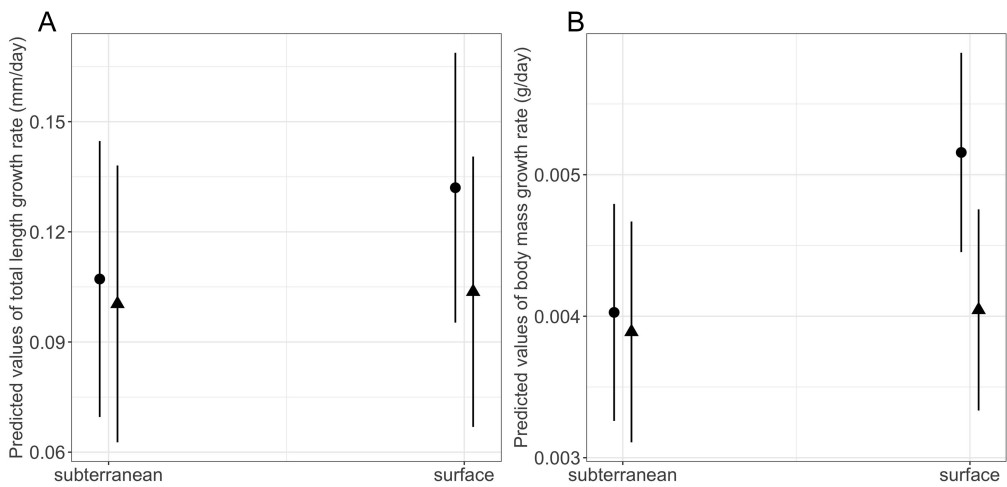

**Figure 6 Interaction effects of habitat type (subterranean *versus* surface) and food availability (high *versus* low) on growth rates in fire salamander (*Salamandra salamandra*) larvae.** (A) Total length growth rate; (B) body mass growth rate (see also Tables 1 and 2 for statistical tests). Closed circles and closed triangles represent the mean values for larvae from high and low food availability treatments, respectively. Lines represent the 95% confidence intervals.

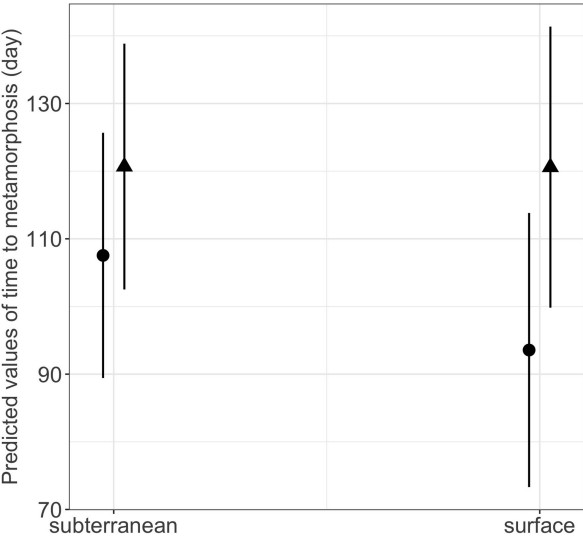

**Figure 7 Interaction effects of habitat type (subterranean *versus* surface) and food availability (high *versus* low) on time to metamorphosis in fire salamander (*Salamandra salamandra*) larvae.** Closed circles and closed triangles represent the mean values for larvae from high and low food availability treatments, respectively (see also Tables 1 and 2 for statistical tests). Lines represent the 95% confidence intervals.

*1973*). In our study, low food availability increased time to metamorphosis in both surface and subterranean salamanders, but only surface larvae displayed enhanced growth rates in high food conditions, suggesting that larvae adapted to high food conditions may capitalize better (*Manenti et al., 2023*). These differences in growth rates among larvae of

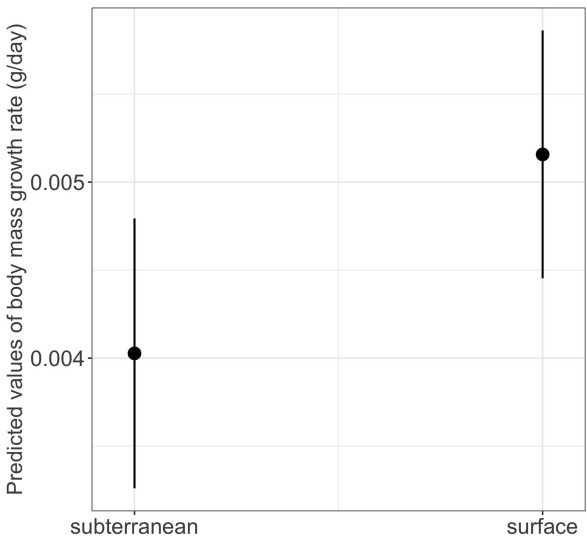

**Figure 8  Effects of habitat type on body mass growth rate in fire salamander (*Salamandra salamandra*) larvae.** Closed circles and lines represent the mean value and the 95% confidence intervals.

different habitat origin suggest an interplay between plasticity and adaptation, favoring the exploitation of contrasting environments in this species. However, it is important to consider the potential carryover effects from conditions experienced at early stages, such as size at metamorphosis and survival at metamorphosis, as these can have significant impacts on the survival and fitness of individuals in their adult life stages (*Melotto, Manenti & Ficetola, 2020*). Our study revealed that the habitat of origin and trophic conditions did not induce differences in size at metamorphosis, suggesting more complex outcomes than those generally supported by the Wilbur-Collins model for amphibian metamorphosis. We showed that fire salamanders can compensate for ecological differences between contrasting habitats during early life stages and, similar to previous research (*Manenti, Denoël & Ficetola, 2013*), our results highlighted their ability to colonise and adapt to new habitats.

The role of the photoperiod in shaping growth and development patterns of amphibian larvae in natural populations is largely unknown. Several physiological studies conducted in the laboratory provided evidence that amphibian larval growth and developmental rates are altered by photoperiod (*e.g.*, *Guyetant, 1964*; *Wright et al., 1988*; *Crawshaw et al., 1992*). These studies, where photoperiod was altered to an extreme degree (*i.e.*, continuous light or darkness), found that amphibians generally increase growth and developmental rates in response to light. However, the response might also be determined by the duration of the photoperiod (*Laurila, Pakkasmaa & Merilä, 2001*). In our study, photoperiod affected only body mass growth rate, which was significantly higher in the 8-hour light treatments.

## Interplay between photoperiod and food availability

A number of studies have documented the impact of foraging shifts interacting with other environmental factors and the induced changes in amphibian growth and development.

These include interactions between foraging shifts and temperature, predation risk and pond desiccation, respectively (*Nicieza, 2000*; *Álvarez & Nicieza, 2002*; *Enriquez-Urzelai et al., 2013*). The results suggested that interactive effects are difficult to predict (*Álvarez & Nicieza, 2002*). As food availability and photoperiod both play a role in growth and development, it is unlikely that they operate independently. Therefore, in our study, we aimed to examine how they interact to affect growth and development rates. Specifically, we hypothesized that photoperiod may mediate the influence of food availability on these rates. Our results did not support this assumption, as the traits we investigated were responsive to food availability and light condition, but no interaction between these two factors was detected. However, a previous study (*Băncilă et al., 2021*) showed that these two factors have an interactive effect on the behavioural responses of fire salamander larvae; more specifically, larvae exposed to low food availability take more risks in exploring the environment, especially in the absence of light, when predator pressure would be lower. *Manenti, Denoël & Ficetola (2013)* found that behavioural plasticity is higher in larvae from subterranean environments. In addition, environmental factors, such as darkness, predator absence, and resource depletion, can interact with other factors, such as prey mobility and conspecific presence, to influence the behavior of cave-dwelling animals (*Uiblein et al., 1992*; *Melotto, Ficetola & Manenti, 2019*). Together, these findings suggest that the adaptive responses to environmental cues are mediated both physiologically and behaviorally during the early development of fire salamanders. As such, in natural conditions, a higher risk-taking behaviour may pay-off in terms of access to resources. We suggest that this, together with lower predation risks, would be an advantage in subterranean environments, even if larval growth rate is slower in the absence of light and/or food resources are limited.

Plasticity may not be selected for in relatively stable environments (*Merilä et al., 2000*; *Kulkarni et al., 2011*), and may not be present in all species because maintaining plasticity is costly (*DeWitt, Sih & Wilson, 1998*; *Jannot, 2009*; *Reed, Schindler & Waples, 2010*). In amphibians, plastic developmental rates have been associated with trade-offs in size at metamorphosis (*Denver, Mirhadi & Phillips, 1998*; *Merilä et al., 2000*). Metamorphosis at a smaller size was associated with reduced adult size and fecundity (*Márquez-García et al., 2009*), lowered immune system function (*Terentyev, 1960*) or increased mortality (*Rudolf & Rödel, 2007*). Larger size at metamorphosis requires additional time for growth, but it is associated with enhanced survival and performance in metamorphs (*Cabrera-Guzmán et al., 2013*). In our study, size at metamorphosis (*i.e.,* total length and body mass) was unaffected by food availability or photoperiod. However, larger larvae (*i.e.,* initial total length) resulted in larger metamorphs (*i.e.,* total length at metamorphosis), while also reaching metamorphosis faster, at slower growth rates, in line with previous studies (*e.g.,* *Alcobendas, Buckley & Tejedo, 2004*). Interestingly, while the initial size at the beginning of the experiment was significantly higher in larvae from subterranean habitats, size at metamorphosis was overall higher in larvae from the surface (although not significantly). This suggests that developmental plasticity can promote development with limited effects on size at metamorphosis.

Plasticity can serve as a potential mechanism that promotes the colonization process, especially in the early stages of facing novel ecological pressures. At this stage, plasticity

allows for flexible responses that can increase the chances of survival and establishment. However, as selective pressures in the new environment persist, adaptation becomes crucial for successful colonization. The fixation of adaptive traits or canalization of plasticity may occur to ensure better fitness in the new environment (*Romero, 2009*; *Levis & Pfennig, 2019*). These mechanisms can coexist in situations where gene flow allows for exchange between populations adapted to the ancestral habitat and populations colonizing the new environment (*Storfer, 1999*; *Richter-Boix et al., 2010*). This idea is consistent with previous studies that have shown how phenotypic plasticity can help species to overcome altered environmental conditions and buy time for population persistence that may be followed by adaptive trait refinement and evolution (*Diamond & Martin, 2021*). However, there are still unanswered questions, such as which are the conditions under which plasticity can allow or limit subsequent evolutionary changes. Subterranean habitats might act as environmental filters and species that occur both in surface and subterranean habitats can be used for comparative analysis with respect to variation of plasticity. Surface habitats, in particular streams, are thought to be the ancestral habitats for fire salamanders (*Seifert, 1991*; *Steinfartz, Weitere & Tautz, 2007*). In the ancestral habitats, more plastic genotypes are expected compared to populations colonizing novel environments (*Diamond & Martin, 2021*). Our results support this prediction, surface larvae showing more plasticity in their growth depending on food availability.

It is important to interpret the results of our study with caution, as the limited number of populations sampled may not fully represent the overall variation in growth rates and time to metamorphosis between surface and subterranean habitats. Specifically, the study only sampled a single surface population, which may limit the generalization of the findings to other surface populations. In this regard, we cannot dismiss the possibility that the genetic component can have a large impact on life-history traits, along with raising conditions (*e.g.*, *Caspers, Steinfartz & Krause, 2015*).

Overall, our study showed that *S. salamandra* is an excellent target species for testing hypotheses regarding populations persistence *via* plasticity and their potential to evolve when confronted with changing environments.

## ACKNOWLEDGEMENTS

We are grateful for the invaluable assistance in the field and laboratory of our friends, family and colleagues: Nicolae Argintaru, Sebastian Topliceanu, Elena Şuşter, Roberto Festuccia, Ştefan Cătălin Baba, Anca Soare, Augustin Nae, Ionuţ Popa, Valerică Toma, Marius Robu, Alexandra Telea, Dragoş Bălăşoiu, Adrian Dumitrescu, Constantin Stănescu and Constantin Stănescu Jr. We also thank three anonymous reviewers and Dr. John Measey for taking the time and effort to review our manuscript; we appreciate all their valuable comments and suggestions, which helped us improve the quality of the paper.

### Funding

This work was supported by two research grants from the Romanian National Authority for Scientific Research, CNCS - UEFISCDI: PN-II-RU-TE-2014-4-1536 and PN-III-P1-1.1-TE-2019-1233. The funders had no role in study design, data collection and analysis, decision to publish, or preparation of the manuscript.

### Grant Disclosures

The following grant information was disclosed by the authors:
The Romanian National Authority for Scientific Research, CNCS - UEFISCDI: PN-II-RU-TE-2014-4-1536, PN-III-P1-1.1-TE-2019-1233.

### Competing Interests

The authors declare there are no competing interests. Dan Cogălniceanu is a member of Chelonia Romania, a non-profit NGO with an academic and nature conservation mission.

### Author Contributions

- Raluca Ioana Băncilă conceived and designed the experiments, performed the experiments, analyzed the data, prepared figures and/or tables, authored or reviewed drafts of the article, and approved the final draft.
- Florina Stănescu conceived and designed the experiments, performed the experiments, analyzed the data, prepared figures and/or tables, authored or reviewed drafts of the article, and approved the final draft.
- Rodica Plăiașu conceived and designed the experiments, performed the experiments, authored or reviewed drafts of the article, and approved the final draft.
- Ioana Nae performed the experiments, authored or reviewed drafts of the article, and approved the final draft.
- Diana Székely conceived and designed the experiments, analyzed the data, prepared figures and/or tables, authored or reviewed drafts of the article, and approved the final draft.
- Sabina E. Vlad performed the experiments, authored or reviewed drafts of the article, and approved the final draft.
- Dan Cogălniceanu conceived and designed the experiments, authored or reviewed drafts of the article, and approved the final draft.

### Animal Ethics

The following information was supplied relating to ethical approvals (i.e., approving body and any reference numbers):

Our research was carried out under permit 78/10.02.2016 from the Speleological Heritage Commission (project no PN-II-RU-TE-2014-4-1536).

### Field Study Permissions

The following information was supplied relating to field study approvals (i.e., approving body and any reference numbers):

Our research was carried out under permit no. 250/20.04.2016 obtained from the Administration of the Domogled-Valea Cernei National Park (project no PN-II-RU-TE-2014-4-1536).

## Data Availability
The raw measurements are available in the Supplementary File.

## Supplemental Information
Supplemental information for this article can be found online at http://dx.doi.org/10.7717/peerj.16046#supplemental-information.

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
