# Peer review of "Food and light availability induce plastic responses in fire salamander larvae from contrasting environments"

_PeerJ, doi:10.7717/peerj.16046_

## Round 0.1 · original submission · Major Revisions

I have now received comments from three reviewers on your manuscript. I have chosen a decision of Major Revisions as all three reviewers have picked up on the ambiguity of sampling locations in the first paragraph of the methods that the reader has.

First, while the sites are given precise coordinates, I agree with reviewer #1 that you would be better to show this detail in a figure of the area - preferably with all water courses and relevant subterranean environments indicated. As mentioned by reviewers #2 & #3, the independence of your sites is important in interpreting your results. Given that you were aware of this in the initial design, I feel that you would be better placed to demonstrate this independence by illustrating your sample design and area in a figure.

Second, given that you state that two different sites were from the same stream, this does suggest that the locations were not independent (Reviewer #2). I agree with Reviewer #3 that if this is indeed the case, then you should make this very clear to the reader throughout your manuscript, especially with your interpretation in the discussion. I also agree with Reviewer #2 that without independence, your statistical approach is invalidated. However, as Reviewer #3 states, as long as this is clear to the reader, and the requisite statements are made in the discussion, the work itself is deserving of publication.

I look forward to receiving a detailed response to all points raised with your revised manuscript (NB I’m not sure why Reviewer #1 had an issue with the figure legends which were clear for me - so no need to change this). If you choose to resubmit, then please consider that your manuscript will be sent to the same reviewers, so I would advise you to comprehensively respond to all of their queries.

Reviewer 1 ·

Basic reporting

The manuscript on food and light availability response in fire salamander is very interesting and provides novel data on this species. The introduction is well written and gives an insight on the research subject. The hypothesis is also clearly defined.
In the results section, figure and table legends are missing (the only legends available are those for tables in supplemental data). Without these, some figures are difficult to interpret.

Experimental design

In the methods section, I suggest adding a map of Romania with marked sites where the samples were collected. From what I saw in google maps, the sites seem to be quite far from each other, so additional information on the climate is also welcomed, since some of the initial differences between populations, might be due to the climate conditions (temperature in the subterranean environment is highly dependent on the average annual temperature of the region), and that can influence the initial size of larvae (e.g. Bergmann's rule), and not necessarily the habitat type (surface/subterranean).

Validity of the findings

It would be useful to reorganize results according to your hypothesis; i.e. it is now difficult to follow which results relate to which part of your hypothesis (and same is true for the discussion)
In the results section, there are many things that need to be revised, and most of them are related to the presentation of the results. First of all, it would be useful to have the initial data on BM and TL for each of the studied populations (summary), as these are important predictors of the size at metamorphosis.
As well, there are some inconsistencies that are very confusing. On several places in the results and the discussion you say that TL0 for subterranean larvae is higher than those from surface (e.g. line 201) and you also say that those that have larger TL0 tend to be larger at metamorphosis (e.g. line 221). And then, you say that that TL at metamorphosis is larger in surface larvae (e.g. line 216). Please, resolve this issue.
In the discussion section, as already mentioned, reorganize the structure to follow the hypothesis, and resolve the issues with the inconsistences on size at metamorphosis and TL0. Furthermore, you use the term “developmental rate” for the first time, and this is something that you do not mention neither in M&M, nor the results sections. Although you maybe refer to growth rate, it is not the same - developmental rate describes larval stage and not size. So, please use consistent terms throughout the manuscript. If you do refer to developmental stage, this should then be added to both, M&M and the results.
In the final paragraph you say that in the ancestral habitats more plastic genotypes should be found and that your data support this prediction. However, it is unclear to what “more” refers to. More plastic genotypes than what? Other species? You do not mention the importance of genotype anywhere in the manuscript. As well, in that context, the differences that you observed could also be due to the differences in the genetics between subterranean and surface populations since they seem to be quite apart geographically. Please refer to this possibility as well.

Additional comments

In the abstract, in the line 40 - add how do they differ in their response.
In the line 239- does “both” refer to both subterranean treatments (high and low food), or this “both” refers to both low food treatments (subterranean and surface)? Please clarify this.
There are also few minor language remarks. In the line 143, do not start a sentence with an abbreviation. In the legend of Table S2 S. salamandra – species name is in capital letter. In Table 2- write BM0 (0 in subscript) instead of BM_0 (and the same refers to TL) to be consistent with the rest of the manuscript.

Reviewer 2 ·

Basic reporting

This well-written manuscript presents the results of a study carried out over two years that investigated the responses of fire salamander larvae that were collected from either a surface stream or two underground caves and raised under a combination of a high or low food treatment in a light or dark environment. The authors expected two-way interactions of the rearing treatments and the habitat of origin on growth and development of fire salamander larvae. They found negative effects of the low food treatment and darkness on several traits but no interaction of food and lighting-treatments. Larvae from caves responded differently to the food treatment compared to surface larvae. These results are interesting given that cave-dwelling fire salamander larvae are thought to be more common than previously expected and hint at the ability of this species to adapt to a variety of habitats. However, there are severe problems with the experimental design (see below).

Experimental design

The authors collected larvae in 2016 (from one stream) and in 2017 (same stream and two different caves), brought them to the cave-lab and raised them under different food and lighting conditions. Initial body mass and size were measured, as well as the time to metamorphosis and growth rate (in length and mass) until metamorphosis. Unfortunately, I have some major issues with the sampling protocol:
1.) larvae were collected from only one surface habitat (over two years). Given that fire salamander females likely show high site-fidelity (e.g., Schulte et al., 2007), many of the larvae might be full or half siblings (even across the two years), and thus do not represent independent replicates. Using site of origin as a random effect does not fix this problem as the independent sample size for stream habitats remains one. Furthermore, treating the second year of stream-sampling (at the same stream) as an independent site is also not valid, given that the same females might have deposited at the same site again. Thus, I recommend including at least one more site of sampling for surface larvae. 2.) Cave larvae were only collected in 2017 and not 2016, thus, the effect of year (even if the experiment was conducted under the exact same conditions) cannot be controlled for. Also, there is no reasoning given in the manuscript why in 2016 only surface larvae were collected. 3.) Initial body size was different among both habitat types. Cave larvae are believed to metamorphose after two years in the water due to the lower water temperatures and lower food availability of the habitat (Manenti et al., 2017). Because age of the larvae could not be controlled for in this study, the time to metamorphosis might actually be drastically different between larvae of the different habitats. I believe that the results could have been more convincing if larvae from the two caves would have also been sampled in 2016.
Manenti, Raoul, Enrico Lunghi, and Gentile Francesco Ficetola. "Cave exploitation by an usual epigean species: a review on the current knowledge on fire salamander breeding in cave." Biogeographia 32.1 (2017): 31-46.
Schulte, Ulrich, Daniel Küsters, and Sebastian Steinfartz. "A PIT tag based analysis of annual movement patterns of adult fire salamanders (Salamandra salamandra) in a Middle European habitat." Amphibia-Reptilia 28.4 (2007): 531-536.

Validity of the findings

Unfortunately, due to the major issues of the sampling protocol (see above), I believe the validity of the findings to be low and recommend this manuscript to be rejected from publication in this journal. I complement the authors on such a nicely written manuscript and suggest to the authors to provide a reasoning on the sampling protocol (see above) or repeat the experiment with more sites before resubmitting to this journal.

Additional comments

lines 118 – 119: Why did sampling of cave-dwelling larvae occur only in 2017?
lines 139 – 140: Was the frequency of feeding events the same?
lines 146 – 147: Wet body mass can be highly variable for a larva due to external factors (e.g., amount of food in the gut or moistness of the skin). How was the larva prepared for measurement? Was it, for example, dried on a piece of paper towel to remove the water on the skin? Was the larva starved for a period of time before taking the weight?
lines 203 – 204: Have larvae been assigned to a treatment randomly? If so, please, mention it in the methods section.

Reviewer 3 ·

Basic reporting

The article is sound and of interest; the background provided and narrative generally work, even if some point should be better addressed and expanded, with the support of the related literature.
English is generally acceptable and understandable (at leats, from a non-mother tongue perspective) while few sentences might need reshaping to lighten their structure and improve clarity
Please, find detailed comments in the attached pdf.

Experimental design

Experimentally, the study was conducted in an approriate way, matching aims and scopes, and with a good methodolical description throughout the method section, with few additional details to be provided. However, I have a big concern regarding the actual availability of replicates for habitat typology, which can potentially undermine the validity, or at least the interpretation (many caveats shoul be discussed), of the findings.
Please, find detail comments in the attached pdf.

Validity of the findings

Results are sound and generally in line with predictions from thoery or other study findings. Statistic works, even though alternative approached may also improve the quality and insightfulness of the findings. Again, the biggest concern is related to the habitat replicate issue (a single surface site, same population).
Conclusion are sound, even if their presentation can be improved, while divergence between different habitats should be more highlighted and stressed (potential local adaptation).
Please, find detailed comments in the attached pdf.

Additional comments

Detailed comments in the attached pdf.

Annotated reviews are not available for download in order to protect the identity of reviewers who chose to remain anonymous.

---

## Round 0.2 · Major Revisions

I'm sending this back to you before I send it out to the reviewers as I'm not happy with the way in which you have dealt with a some of the comments.

The major issue with your previous submission was the confusing description of the sites. Now we have a screenshot of a map (inadequate), and there are now 4 site names but you clearly indicate 3 populations. But on the map, two are (probably) v. close together. This is not good enough and I won't send it out for review again until you've fixed this. (1) Please provide a figure for the main manuscript that comprehensively helps the author understand your sampling. (2) Improve the text so that it is less confusing with respect to the number of sites and sampling occasions. (3) As indicated in my decision, please include your experimental design as a part in this figure (composite figure)

Reviewer #1 pointed out that it would be good to have a summary of initial data on BM and TL for each of the studied populations. Your response was to add a figure in the Supplemental Info and point out that raw data is available. Please follow the instruction to include a summary. The Supp Info figure suggests some problems with initial size of these animals. You do need to be transparent about any issues that you had at the start of the experiment. It seems to me that the next issue pointed out by Rev#1 is because there were significantly different starting points for these two treatments.

Reviewer #1 also asked that you re-organise your results according to the hypotheses posed. Your response is that you have done this, but for me, I can't follow the results and answer the hypotheses with ease. Use subheadings for each hypothesis and then provide only those results that respond to that specific hypothesis under each subheading.

Please make sure that you address every point raised within the manuscript before resubmitting. I can't send this out for review again unless I feel that it would not waste the time of my reviewers. As long as I can't easily understand the manuscript, it will be sent back to you. I hope that's clear.

---

## Round 0.3 · Minor Revisions

Thanks for your revised manuscript. There are some minor corrections needed before this can be accepted. Reviewer 1 has provided some comments and Reviewer 3 has provided some comprehensive comments (on an attached file). I look forward to accepting your revision once these minor revisions are made. Please note comments below in your legends.

Figure 1 – thank you for this revision which is very welcome. The figure is now clear, however the legend is insufficient. Legend text needs to be standalone (your other legends are much better). Is the white line a political boundary (needed?) – needs explanation. Software details not needed, just data source.
Figures 2 & 3 – please could you add data points to these graphics?
All figures – I suggest that you add the Latin name
Tables 1 & 2 – legends don’t mention what is being measured.

Acknowledgements – your reviewers have really been very helpful. It would be good to see your appreciation reflected here.

Reviewer 1 ·

Basic reporting

The corrected version of the manuscript on food and light availability response in fire salamander is significantly improved in comparison to the first version and is now much easier to follow.

Experimental design

The authors have explained all the ambiguities that were present in the first version of the manuscript and now the experimental design sounds good.

Validity of the findings

The authors have significantly improved the manuscript and as it is now presented, the results conform to the findings of the manuscript.

Additional comments

There are few minor linguistic corrections that I suggest to be made prior to the final acceptance of the manuscript. Specific comments are written below:
Line 181 – please correct "There was no mortality in OUR study" (not your)
Line 309 – maybe instead of "The study" at the beginning of the sentence, write „the aforementioned study“ or „their study“ since „THE study“ could refer to both, study from Manenti et al, or to your study
Line 359 – instead of „Our results did not support this assumption.“ I suggest merging it with the sentence before, and adding „but“ between these two sentences in order to stress that your results showed something you didn't expect

Reviewer 3 ·

Basic reporting

The manuscript has well improved. See the attached document for specific comments

Experimental design

Sound, as long limitations deriving from single surface population sampling are well clear

Validity of the findings

same, as long limitations deriving from single surface population sampling are well clear. Discussion needs to be improved in few points.

Annotated reviews are not available for download in order to protect the identity of reviewers who chose to remain anonymous.

---

## Round 0.4 · accepted · Accept

Thanks for your corrections. I am now happy to accept this manuscript for publication.